# What will the cardiovascular disease slowdown cost? Modelling the impact of CVD trends on dementia, disability, and economic costs in England and Wales from 2020–2029

Brendan Collins[1]*, Piotr Bandosz[2], Maria Guzman-Castillo[3], Jonathan Pearson-Stuttard[4], George Stoye[5], Jeremy McCauley[6], Sara Ahmadi-Abhari[4], Marzieh Araghi[4], Martin J. Shipley[7], Simon Capewell[1], Eric French[8], Eric J. Brunner[7], Martin O'Flaherty[1]

1 Department of Public Health, Policy and Systems, University of Liverpool, Liverpool, United Kingdom, 2 Medical University of Gdansk, Gdansk, Poland, 3 Department of Social Research, University of Helsinki, Helsinki, Finland, 4 School of Public Health, Imperial College London, London, United Kingdom, 5 Institute for Fiscal Studies, London, United Kingdom, 6 School of Economics, University of Bristol, Bristol, United Kingdom, 7 Institute of Epidemiology & Health, University College London, London, United Kingdom, 8 Faculty of Economics, University of Cambridge, Cambridge, United Kingdom

* brenc@liv.ac.uk

**Data Availability Statement:** Data to run the model are included in the supplementary materials.

## Abstract

### Background

There is uncertainty around the health impact and economic costs of the recent slowing of the historical decline in cardiovascular disease (CVD) incidence and the future impact on dementia and disability.

### Methods

Previously validated IMPACT Better Ageing Markov model for England and Wales, integrating English Longitudinal Study of Ageing (ELSA) data for 17,906 ELSA participants followed from 1998 to 2012, linked to NHS Hospital Episode Statistics. Counterfactual design comparing two scenarios: **Scenario 1. CVD Plateau**—age-specific CVD incidence remains at 2011 levels, thus continuing recent trends. **Scenario 2. CVD Fall**—age-specific CVD incidence goes on declining, following longer-term trends. The main outcome measures were age-related healthcare costs, social care costs, opportunity costs of informal care, and quality adjusted life years (valued at £60,000 per QALY).

### Findings

The total 10 year cumulative incremental net monetary cost associated with a persistent plateauing of CVD would be approximately £54 billion (95% uncertainty interval £14.3-£96.2 billion), made up of some £13 billion (£8.8-£16.7 billion) healthcare costs, £1.5 billion (-£0.9-£4.0 billion) social care costs, £8 billion (£3.4-£12.8 billion) informal care and £32 billion (£0.3-£67.6 billion) value of lost QALYs.

**Funding:** British Heart Foundation RG/16/11/32334. The funder provided support in the form of salaries for authors [BC, JPS, MGC, PB, SA, MA, GS, JM], but did not have any additional role in the study design, data collection and analysis, decision to publish, or preparation of the manuscript. The specific roles of these authors are articulated in the 'author contributions' section.

**Competing interests:** All authors have completed the Unified Competing Interest form (available on request from the corresponding author) and declare: no support from any organisation for the submitted work; no financial relationships with any organisations that might have an interest in the submitted work in the previous three years. Dr Collins is currently on secondment as Head of Health Economics in Welsh Government; this paper does not represent any views of Welsh Government. Dr Pearson-Stuttard is also Head of Health Analytics at a commercial company, Lane Clark & Peacock LLP, vice-chair of the Royal Society for Public Health and reports personal fees from Novo Nordisk A/S, all outside of the submitted work. This work was completed as part of Dr Pearson-Stuttard's academic appointment at Imperial College London.

## Interpretation

After previous, dramatic falls, CVD incidence has recently plateaued. That slowdown could substantially increase health and social care costs over the next ten years. Healthcare costs are likely to increase more than social care costs in absolute terms, but social care costs will increase more in relative terms. Given the links between COVID-19 and cardiovascular health, effective cardiovascular prevention policies need to be revitalised urgently.

## Introduction

The dramatic declines in cardiovascular disease (CVD) mortality in high income countries were a great success story of the late 20[th] century. However, since about 2011, that fall in CVD mortality stalled, with consequent slowing of improvements in life expectancy in England, Wales, United States [1] and elsewhere [2, 3]. This plateauing of CVD mortality appears to be mainly due to slowing in CVD incidence declines, rather than slowing in case fatality improvements among patients treated for CVD [4]. The underlying reasons for the mortality slowdown are disputed. However, adverse risk factor trends in obesity and type 2 diabetes [5, 6], may now cancel out the benefit of declines in smoking and hypertension prevalence. Furthermore, although these trends have been repeatedly documented, their overall effects on health care spending are uncertain.

Reducing CVD and dementia incidence are key goals of the English National Health Service [7]. Because smoking, diabetes, hypertension and obesity are shared risk factors for CVD and dementia, trends in their incidence are strongly related. Furthermore, CVD is a major risk factor for disability as well as dementia, and thus impacts both health care costs and social care costs [4, 8]. However, whether these changes in disease incidence will increase or decrease future health and social care costs is unclear. Increased disease incidence might raise survivor numbers and associated costs; but, conversely, increased mortality might reduce prevalence so that fewer people require care [9]. However, understanding how these costs evolve will be crucial for healthcare planning.

This paper thus aims to fill a key gap in the literature. by modelling and forecasting the health impact and economic costs of the recent slowing of the decline in CVD incidence in England and Wales. We have therefore linked individual-level health care and social care costs for participants in the English Longitudinal Study of Ageing (ELSA) [10]. Health care cost estimates have long existed according to disease [11]; however disaggregated social care cost estimates have only recently become available. We use a probabilistic health transition Markov model to forecast trends in diseases and how these epidemiological trends are likely to impact future spending. Our objective is to estimate inclusive economic costs, health and social care costs and quality-adjusted life years (QALYs) for the population in England and Wales from 2020 to 2029, and to estimate costs specifically attributable to CVD and dementia, as a consequence of the recent plateauing in CVD incidence rate. We have therefore compared two scenarios: 1. Assuming age-specific CVD incidence remains plateaued at 2011 levels, (continuing recent trends), or Scenario 2. Assuming age-specific CVD incidence continues to decline, following the longer-term trends, but only likely if CVD prevention policies are re-energised.

## Methods

IMPACT BAM's epidemiological methods have been validated and reported in detail previously [4, 12]. In this study we therefore focus on the additional economic developments.

**Table 1. Summary of model inputs, with data sources, regression methods and distributions.** Full regression equations are shown in Appendices 1 and 2 in S1 File.

| Model input | Source | Regression methods | Distributions |
|---|---|---|---|
| Prevalence of initial states | ELSA data fitted using curve fitting tool in MATLAB | N/A | |
| Transition probabilities | ELSA data | Logistic regression | |
| Healthcare costs | | | |
| Hospital costs | ELSA-HES linkage | OLS regression | Beta with +/- 20% |
| Prescribing costs | ELSA combined with BNF | Two part Probit + OLS regression | Beta with +/- 20% |
| Social care costs (Cleaner/Homecare/Daycare) | ELSA data combined with PSSRU Reference costs | Two part Probit + OLS regression | Beta with +/- 20% |
| Residential care costs | ELSA data combined with PSSRU Reference costs | Probit | Beta with +/- 20% |
| Informal care costs | ELSA data combined with ONS GVA data | Two part Probit + OLS regression | Beta with +/- 20% |
| Utility index (EQ-5D) values | ELSA data combined with UK reference values from Janssen & Szende (2014), disease multipliers from Sullivan (2011) and Health Survey for England data on ADL deficits and EQ-5D index values. | Linear regression of HSfE | Fixed values only |

Table 1 lists the model inputs which are explained in more detail in Appendices 1 and 2 in S1 File.

We used a simulation modelling approach to forecast future healthcare, social care and informal care costs, and QALYs across the population under two diverging scenarios of future CVD incidence.

Simulations of health transitions for people aged 35–100 in England and Wales were carried out using the previously validated IMPACT Better Aging Model (BAM). This open-cohort, stochastic Markov model synthesises observed trends in incidence of CVD, dementia, disability and mortality, based on data from the English Longitudinal Study of Ageing (ELSA) [10] and Office for National Statistics (ONS). Model inputs for Wales were estimated using English ELSA and National Health Service Hospital Episode Statistics (HES) data, and ONS data that included Wales. The IMPACT BAM model uses ELSA data for information on health transition probabilities, and projects to the future using ONS demographic and mortality data.

## Transition probabilities

Transition probabilities were obtained as a function of age and sex from incident cases between wave n and n+1 in ELSA. As with estimates of prevalence values, the transition probabilities obtained from pooling ELSA epochs were attributed to the mid-point of the data collection period. A new cohort of those reaching age 35 each year enters through the disease-free state (see model figure in Appendix in S1 File). The prevalence of cardiovascular disease and functional impairment is very low in this 35-year old cohort (<2% in total) therefore, the resulting error in misclassification is negligible. Movements between states occur every year in the model based on transition probabilities. The transition probabilities between states were calculated using logistic regression of 2-year incidence of CVD, cognitive impairment, functional impairment, and recovery from functional impairment, using the ELSA data with age, sex and current health state as coefficients (where dementia was classed as concurrent cognitive and functional impairment). A calendar effect was added where CVD incidence trends mirrored CVD mortality trends and cognitive impairment was set to decrease by 2.7% per year, based on trends in the ELSA data.

Probability of death was estimated using a three step model; for the first step, CVD and non-CVD mortality probabilities of CVD up 2025 in 5-year age bands were calculated using

the Bayesian Age Period Cohort (BAPC) model, with ONS mortality and population estimates from 1982–2012 for England and Wales as inputs. In the second step, we calculated mortality rates from ELSA for the age groups 50–59, 60–69, 70–79, and 80–89 and fitted two logistic regression models, first including only sex, gender, and interactions, and secondly also including the model health state. The results of the first step gave probability of CVD and non-CVD death by sex, single year of age and calendar year, which were adjusted by the results of the second step to also give probability of CVD and non-CVD death by health state in the model. These methods were chosen to favour the population-level data from ONS but adjusting for the ELSA-specific data to estimate mortality risk by health state in the model.

The synthesised trends were projected from 2011 to 2029 (with outcomes measured for the remaining ten years of the NHS Plan from 2020–29) [13] based on trend data from ELSA waves 1–6 (2002/03 to 2012/13) and mortality trends from 1990–2016. There have been subsequent ELSA waves but these ones are the only ones that have been linked with resource use data. IMPACT BAM has eight health states: free of cardiovascular disease (CVD), cognitive impairment (CI) or functional impairment (FI); CVD; CVD and FI; CVD and CI; CVD and dementia; dementia; CI and FI; and two additional absorbing states of CVD death and non-CVD death. The main model outcomes were health and social care costs, value of informal care, and QALYs experienced. QALYs were valued at £60,000 based on UK Treasury Green Book [14] but with a sensitivity analysis using £30,000 which is often quoted as the threshold used by the National Institute for Health and Care Excellence (NICE).

## Healthcare costs

Healthcare costs included all recorded costs for individuals, so were not specific to CVD and dementia, and include future healthcare costs, which is why the model is useful for answering whether preventing CVD saves costs in the longer term if it increases survival. The ELSA data was linked with Hospital Episode Statistics (HES) data (inpatient, outpatient, A&E) which was costed using NHS Healthcare Resource Groups (HRGs) for 2018/19 financial year. 80% of ELSA participants (14,789 of 18,529) gave consent for their records to be linked. Ordinary Least Squares (OLS) regression using data from the consenting sample was used to estimate the relationship between total hospital costs and health state, age and gender. These regression results were used to impute costs for those also in the non-consenting sample. ELSA respondents provide information on prescribed medications currently being taken. For each prescription, we use the British National Formulary (BNF) paragraph, section, and chapter number, and match this to the Net Ingredient Costs (NICs) contained in Prescription Cost Analysis compiled by the NHS Health and Social Care Information Centre. Total healthcare costs were then calibrated to estimates of total healthcare costs by age for the UK reported by the Office for Budget Responsibility [15] to account for missing costs like primary care, community, and other underreporting of costs in the linked ELSA-HES data. We do not assume any changes in costs over time due to new technologies, price or wage inflation, or other causes, so the modelling assumes that costs for each health state remain constant over time.

## Social care and informal care costs

Age-related social care costs were estimated using reported social care contact hours from ELSA combined with Personal Social Service Research Unit (PSSRU) unit costs [16]. These were for five resources; cleaner, care/nursing home staff, other formal help, Local Authority-provided home care worker/ home help, and non-Local Authority home care worker/ home help. We added residential care costs to IMPACT-BAM from a logit regression of whether the ELSA member is currently living in institutional care with a representative sample of people

living in institutions, but we ran the institutional care costs regressions by having the dependant variable as dummy of whether they were living in an institution at the time of the interview. We then assumed an average yearly cost of living in institutional care of £39,156 based on PSSRU reference costs (This is based on a 50:50 split between residential and nursing beds). The social care resource use does not include some costs such as costs of home improvements, and respite care. To account for the many ELSA respondents who report zero hours of social care use, we estimated a two-part model: (i) a probit for the presence of any social care use, and (ii) OLS regression for the amount of resource use for people who report non-zero hours of social care receipt. For daycare, data on receipt of daycare are available but not number of hours. We therefore used a probit model and applied an average annual cost from PSSRU reference costs (£7,280 in 2016/17 prices). The PSSRU estimates do not contain recent estimates for the costs of cleaners so we assumed an hourly cost of 1.5 times the national living wage.

We calculated informal care based on the number of hours of help that ELSA respondents reported they had received in the last week from up to 25 different people, ranging from spouses to neighbours. We assumed an average cost per hour of informal help of £7.76, based on data from ONS on gross value added of informal care less household inputs, and total number of hours of informal care. All health, social and informal care costs were inflated or deflated to 2019 prices using Treasury GDP Deflator (October 2018). The modelling was undertaken in real terms (i.e. in current prices) in line with the suggestion of the UK Treasury Green Book but with an additional sensitivity analysis where costs were discounted at 3.5% per annum. For probabilistic sensitivity analyses, costs were fitted to a beta distribution where the 95% uncertainty intervals represented +/- 20% of the median, which was applied in addition to epidemiological uncertainty around the proportion of the population in each health state in the model. The modelled costs for England were used as inputs for the England and Wales population in IMPACT BAM.

## Calculating quality adjusted life years

Utility weights for QALYs were taken from the EQ-5D MEPS (Medical Expenditure Panel Survey) catalogue [17] and Health Survey for England [18]. Based on a linear regression of Health Survey for England 2012 (the most recent year that included all of these variables), the coefficient of EQ-5D index score for number of limitations in activities of daily living (ADLs) was -0.042 after controlling for health, age and gender. For each health state/age combination, we multiplied the population norm EQ-5D index score from Jannsen and Tzende [19] by an EQ-5D multiplier for cognitive impairment, dementia, or CVD (from the MEPS) and by the ADL decrements for the distribution of number of ADLs in that state.

QALYs were not discounted in the main scenario, but an additional scenario has QALYs discounted by 1.5% per annum and by 3.5% per annum, in line with NICE and UK Treasury Green Book. There was no probabilistic distribution added to the QALY weights because the uncertainty on QALY weights is very low, so any difference in QALYs in the results is driven only by epidemiological uncertainty.

Please see Technical Appendix for further details of our economic methods.

## Scenarios modelled

We modelled undiscounted health and social care costs and QALYs for 2020–2029 under two scenarios:

**Scenario 1. CVD Plateau**–Assuming age-specific CVD incidence remains at 2011 levels, continuing recent trends.

**Scenario 2. CVD Fall**—Assuming age-specific CVD incidence continued to decline, following the long-term trends from 1991 to 2011.

The CVD plateau is the most likely of these two scenarios which were selected to give a comparison of the potential future trajectory for CVD trends. There are several countries such as Spain and France [20] where CVD has continued to decline beyond what has been achieved in England and Wales–although improvements have slowed across Europe—so it was felt that this comparison would be useful in understanding the costs of the slowdown in CVD improvements and the potential economic value of improvements that might be achieved in the NHS plan for England, if it was to produce a return to an improvement of the CVD trajectory. Fig 1 shows what the two scenarios mean in terms of CVD incidence, prevalence and mortality trends.

### Forecasting future costs

We calculated total costs for the whole England and Wales population as well as the specific excess costs of dementia and CVD. We estimated specific excess costs of dementia and CVD by comparing costs of individuals with dementia or CVD with the costs of individuals who were identical in age, gender and other disabilities who did not have dementia or CVD. Dementia was defined as the presence of both cognitive and functional impairment and the excess costs of dementia were estimated by comparing the same people as if they only had functional impairment.

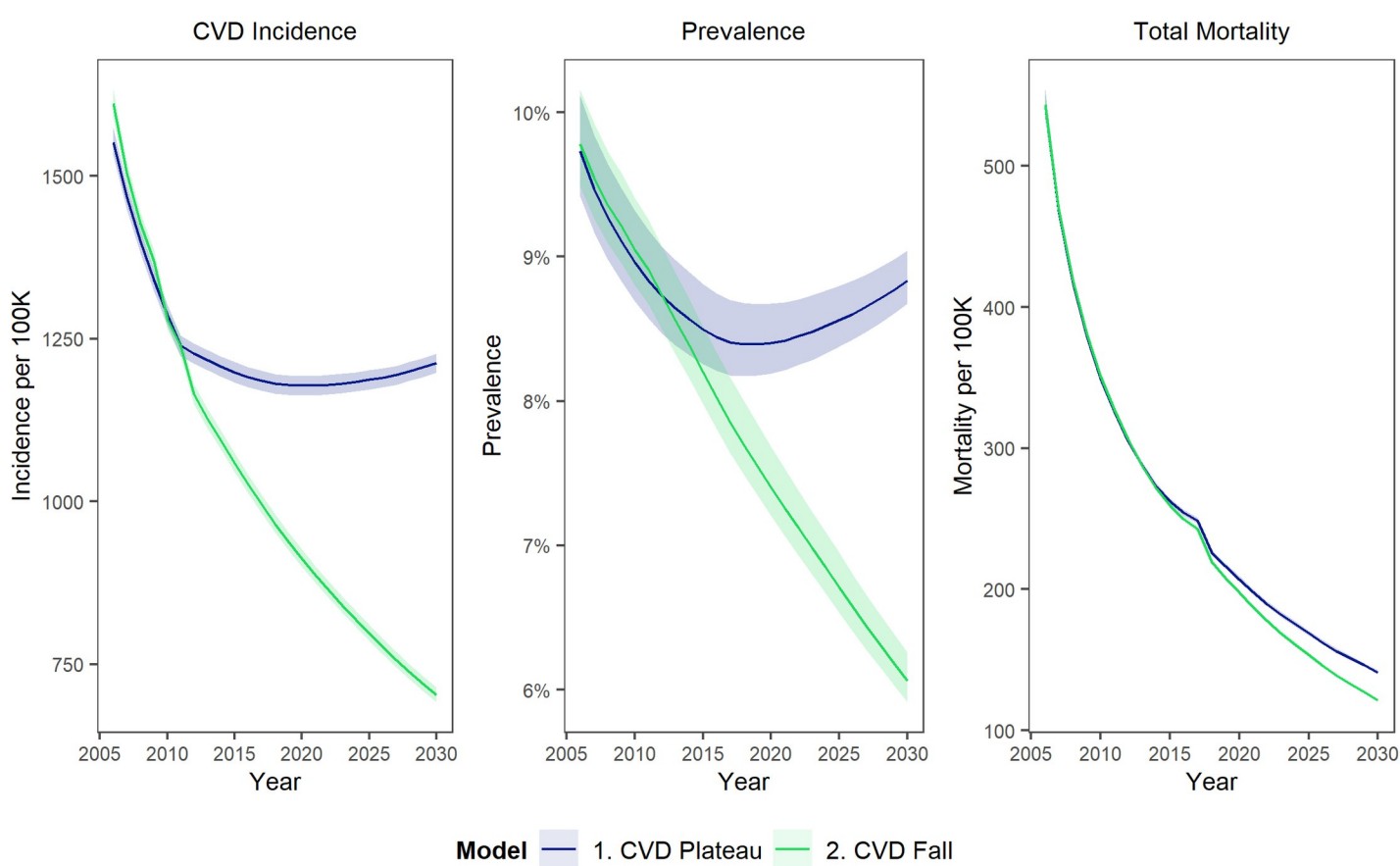

**Fig 1. Modelled CVD incidence per 100,000 population aged 35–100, prevalence (% of people aged 35–100), and mortality per 100,000 population aged 35–100, from 2005 to 2030, comparing Scenario 2 (Continuing decline in CVD incidence) with Scenario 1 (plateaued CVD incidence).**

**Table 2. Total cost of illness for CVD and dementia in 2020.** £billions (2019 prices).

| Disease | Healthcare | Social care | Value of informal care | Value of Disease-Related QALYs lost | Total value of healthcare costs and QALY losses |
|---|---|---|---|---|---|
| CVD | | | | | |
| All ages | 5.29 (4.17 to 6.37) | 1.03 (0.81 to 1.24) | 3.25 (2.56 to 3.91) | 6.51 (6.77 to 6.32) | 16.08 (14.30 to 17.83) |
| Age 35–64 | 1.97 (1.53 to 2.41) | 0.11 (0.08 to 0.13) | 0.98 (0.77 to 1.21) | 2.53 (2.79 to 2.35) | 5.59 (5.17 to 6.11) |
| Age 65–79 | 2.18 (1.72 to 2.61) | 0.34 (0.27 to 0.41) | 1.32 (1.04 to 1.58) | 2.65 (2.70 to 2.61) | 6.50 (5.73 to 7.22) |
| Age 80–100 | 1.14 (0.90 to 1.37) | 0.58 (0.46 to 0.70) | 0.95 (0.75 to 1.14) | 1.32 (1.34 to 1.30) | 4.00 (3.45 to 4.52) |
| Dementia | | | | | |
| All ages | 1.71 (1.34 to 2.06) | 5.06 (3.97 to 6.06) | 3.51 (2.74 to 4.23) | 4.20 (4.45 to 3.97) | 14.48 (12.49 to 16.32) |
| Age 35–64 | 0.21 (0.15 to 0.28) | 0.28 (0.20 to 0.37) | 0.38 (0.27 to 0.52) | 0.55 (0.69 to 0.43) | 1.42 (1.32 to 1.61) |
| Age 65–79 | 0.85 (0.66 to 1.02) | 1.62 (1.26 to 1.96) | 1.60 (1.25 to 1.94) | 2.08 (2.24 to 1.94) | 6.15 (5.40 to 6.86) |
| Age 80–100 | 0.66 (0.52 to 0.79) | 3.15 (2.48 to 3.79) | 1.51 (1.19 to 1.82) | 1.56 (1.62 to 1.51) | 6.89 (5.81 to 7.90) |

(95% uncertainty intervals in brackets). Results shown are from scenario 2 (CVD fall)–however results are broadly similar for both scenarios.

Healthcare costs are total NHS costs based on ELSA data linked with NHS England HES data. Social care costs are based on ELSA and include cleaner, care/nursing home staff, other formal help, Local Authority-provided home care worker/ home help, and non-Local Authority home care worker/ home help, as well as residential care. Informal care costs are based ELSA data multiplied by ONS estimates of gross value added per hour of care. QALYs are quality adjusted life years and are valued at £60,000 per QALY. Note that QALYs reflect only the uncertainty in the epidemiology, not uncertainty around the QALY impacts of disease, which is reflected in very tight uncertainty intervals.

## Results

A continuing CVD plateau (Scenario 1) would mean that annual CVD incidence remains at around 1,200 per 100,000 people aged 35–100. CVD prevalence would increase slightly over time to around 9% of 35–100 year olds in 2029, reflecting demographic aging (Fig 1). Conversely, a further fall in CVD (Scenario 2)--would see CVD incidence decline to below 800 per 100,000. CVD prevalence would correspondingly fall to approximately 6% of 35–100 year olds (Fig 1).

Tables 2 and 3 show the healthcare costs per year from scenario 2 (CVD Fall)–in practice these are very similar in scenario 1 as well as they do not vary with changes in prevalence. The model suggests that in 2020 (the base year), total healthcare costs were approximately £5.3billion for CVD and some £1.7billion for dementia, while total social care costs were approximately £1billion for CVD and £5billion for dementia. The value of informal care was approximately £3.2billion for CVD and £3.5billion for dementia in 2020 (Table 2).

Compared with Scenario 2, a plateaued CVD incidence (Scenario 1) could result in approximately 16% higher average healthcare costs from 2020 to 2029, 1.1% higher social care costs, 2.8% higher costs of informal care, and 0.2% fewer QALYs experienced (Fig 2).

Excess healthcare costs of CVD and dementia per person-year (the costs compared to a counterfactual where an individual in the same age group did not have CVD, or dementia) would be similar across age groups (approximately £2,300 for CVD and £4,200 for dementia). However, social care costs would increase across age groups from approximately £130 per person in 35–64 year olds to around £1,155 per person-year in 80–100 year olds for CVD, and from approximately £5,763 to £19,913 for dementia. Informal care costs would increase from

**Table 3. Excess cost (£, 2019 prices) per person, per year with CVD and dementia, 2020 (compared to if the same people did not have CVD and/or dementia).**

| Disease | Healthcare | Social care | Value of informal care | Value of Disease-Related QALYs lost | Total value of healthcare costs and QALY losses |
|---|---|---|---|---|---|
| CVD | | | | | |
| All ages | 2,330 (1,840 to 2,790) | 454 (357 to 545) | 1,433 (1,132 to 1,717) | 2,868 (2,862 to 2,876) | 7,087 (6,197 to 7,916) |
| Age 35–64 | 2,395 (1,891 to 2,869) | 130 (102 to 158) | 1,196 (943 to 1,434) | 3,094 (3,090 to 3,099) | 6,813 (6,029 to 7,552) |
| Age 65–79 | 2,309 (1,823 to 2,764) | 362 (284 to 436) | 1,395 (1,099 to 1,674) | 2,806 (2,803 to 2,809) | 6,868 (6,008 to 7,677) |
| Age 80–100 | 2,265 (1,788 to 2,711) | 1,155 (907 to 1,389) | 1,890 (1,487 to 2,262) | 2,618 (2,616 to 2,621) | 7,928 (6,799 to 8,978) |
| Dementia | | | | | |
| All ages | 4,209 (3,323 to 5,041) | 12,417 (9,843 to 14,944) | 8,626 (6,828 to 10,332) | 10,360 (10,324 to 10,398) | 35,602 (30,368 to 40,652) |
| Age 35–64 | 4,282 (3,382 to 5,139) | 5,763 (4,558 to 7,007) | 7,976 (6,264 to 9,598) | 11,404 (11,383 to 11,427) | 29,411 (25,566 to 33,137) |
| Age 65–79 | 4,229 (3,339 to 5,071) | 8,137 (6,456 to 9,762) | 8,044 (6,375 to 9,614) | 10,493 (10,473 to 10,513) | 30,901 (26,684 to 34,917) |
| Age 80–100 | 4,159 (3,282 to 4,981) | 19,913 (15,720 to 23,863) | 9,579 (7,556 to 11,448) | 9,873 (9,872 to 9,874) | 43,535 (36,441 to 50,132) |

Results shown are from scenario 2 (CVD fall)–however results are broadly similar for both scenarios. Note some people have both CVD and dementia.

Data sources same as Table 1. QALYs are quality adjusted life years and are valued at £60,000 per QALY.

approximately £1,196 in 35–64 year olds to £1,890 for 80–100 year olds for CVD and increase from approximately £7,976 in 35–64 year olds to around £9,579 in 80–100 years old with dementia (Table 3).

Table 4 shows the cumulative costs of CVD, dementia and total costs across the whole population (including people without CVD or dementia) for the two scenarios. Compared with Scenario 2, the plateau in CVD mortality since 2011 (Scenario 1) is projected to produce a cumulative net monetary cost of around £54 billion (95% uncertainty interval £14.3-£96.2 billion), made up of approximately £13 billion (£8.8-£16.7 billion) healthcare costs, £1.5 billion (-£0.9-£4.0 billion) social care costs, £8 billion (£3.4-£12.8 billion) informal care and £32 billion (£0.3-£67.6 billion) value of lost QALYs in the ten years from 2020 to 2029. Of these costs, cumulative CVD-specific costs (including value of QALYs lost) are projected to be approximately £39 billion higher with the CVD plateau, whereas dementia-specific costs are projected to be actually slightly lower (£0.6billion), reflecting fewer patients surviving to old age. Sensitivity analyses with 1.5% (QALYs) and 3.5% (costs) discount rates, and lower QALY valuations of £30,000 per QALY, are shown in Appendix 3 in S1 File. Using discounted instead of undiscounted costs and QALYs only slightly reduced the difference between the scenarios, while valuing QALYs at £30,000 reduced the net monetary cost difference between the scenarios to around £37.7billion.

## Discussion

### Summary of results

After previous, dramatic falls, CVD incidence and mortality have recently plateaued in the UK. This slowdown could substantially increase health and social care costs over the next ten years, and cumulatively cost approximately £54 billion by 2029. The additional £22 billion in health, social and informal care costs, would represent about a 1.6% increased demand on

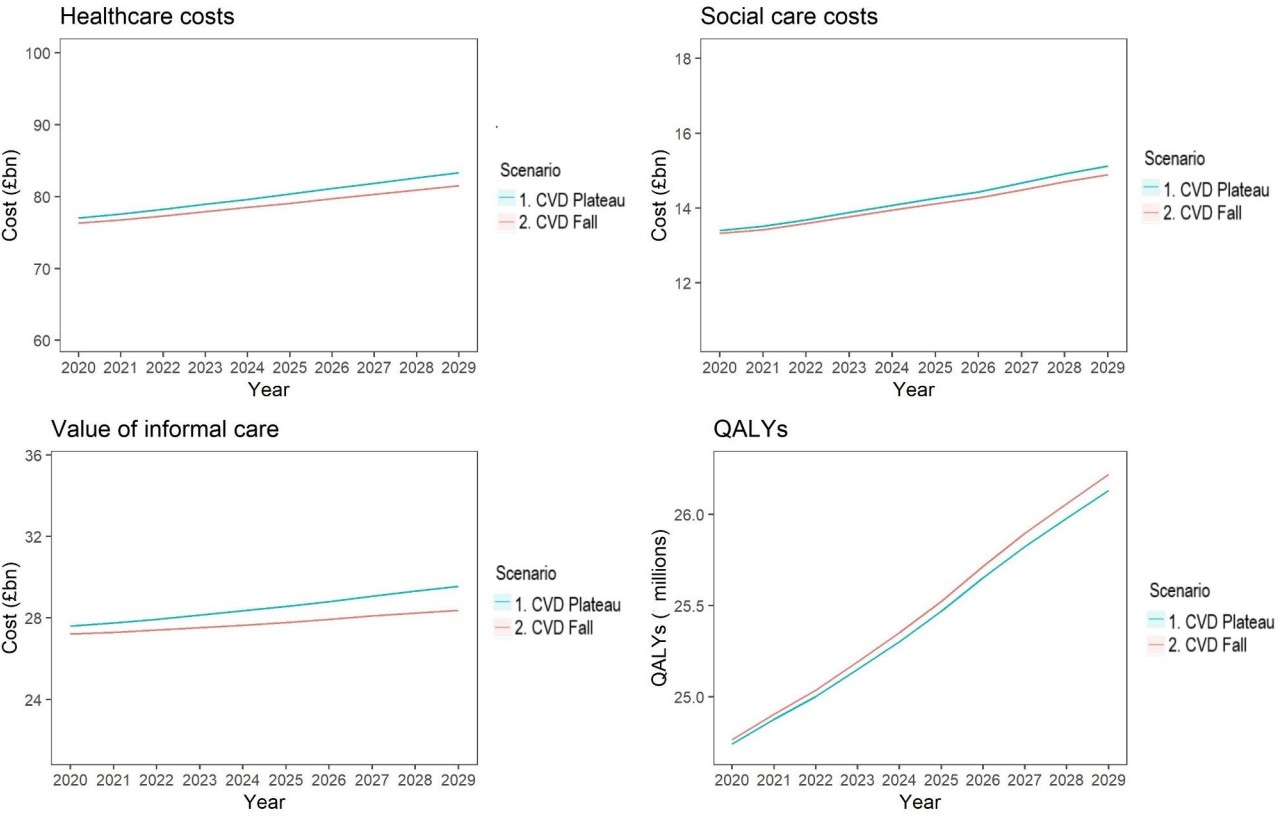

**Fig 2. Modelled healthcare costs, social care costs, value of informal care, and QALYs, from 2020–2029, comparing Scenario 2 (Continuing decline in CVD incidence) with Scenario 1 (plateaued CVD incidence).** Healthcare costs are total NHS costs based on ELSA data linked with NHS England HES data. Social care costs are based on ELSA and include cleaner, care/nursing home staff, other formal help, Local Authority-provided home care worker/ home help, and non-Local Authority home care worker/ home help, as well as residential care. Informal care costs are based ELSA data multiplied by ONS estimates of gross value added per hour of care. QALYs are quality adjusted life years experienced per year, across the whole population, aged 35–100.

NHS and social care budgets, which are already strained. However, the biggest costs would be approximately 540,000 lost QALYs (reflecting worse quality-of-life from higher rates of CVD and disability, and more life-years being lost through increased mortality). The immediate impact would particularly hurt the NHS, with a more distal and delayed impact on informal care and social care.

## Comparison with other studies

Our results generally endorse and expand on previous studies of CVD and dementia costs. The MODEM study estimated the total (not excess) costs of people with dementia in England to be approximately £24 billion in 2015, made up of £10 billion unpaid care, £10 billion social care and £4 billion in health care costs [21]. Healthcare costs per person per year were £3,025 for people with mild dementia, up to £4,800 for severe dementia, and £4,800 for all care home residents with dementia. These costs are similar to our excess healthcare costs of approximately £4,400 for dementia. Luengo Fernandez et al. [22] reported a similar figure of 17,000 Euros for the combined health and social care cost per dementia patient for the UK in 2007.

Total healthcare costs in our study for people aged 35 and over were around £80billion per year, similar to NHS data in 2017/18 showing costs of approximately £108 billion in England and £7billion for Wales for all ages [23].

**Table 4. Total cumulative undiscounted health and social care costs, value of informal care, and value of QALYs (where 1 QALY valued at £60,000) for adults aged 35–100 in England and Wales, over 10 years from 2020–2029.**

| Population | Scenario | Healthcare | Social care | Value of informal care | Total costs | Value of QALYs lost (billions) |
|---|---|---|---|---|---|---|
| | Scenario 1 | 62.9 | 12.1 | 38.3 | 113.2 | 77.2 |
| | | (49.8 to 75.5) | (9.5 to 14.5) | (30.3 to 46.0) | (89.6 to 136.0) | (79.5 to 75.5) |
| **CVD** | Scenario 2 | 49.7 | 10 | 30.9 | 90.6 | 61.2 |
| | | (39.2 to 60.0) | (7.8 to 12.0) | (24.3 to 37.1) | (71.2 to 109.0) | (63.5 to 59.5) |
| | Difference (1–2) | 13.1 | 2.1 | 7.4 | 22.6 | 16.1 |
| | | (9.8 to 16.7) | (1.6 to 2.7) | (5.6 to 9.4) | (17.0 to 28.7) | (13.1 to 19.0) |
| | Scenario 1 | 17.4 | 52.7 | 36 | 106.1 | 42.8 |
| | | (13.8 to 21.1) | (41.7 to 63.4) | (28.6 to 43.5) | (84.1 to 128.1) | (45.3 to 40.4) |
| **Dementia** | Scenario 2 | 17.7 | 52.5 | 36.1 | 106.3 | 43.1 |
| | | (13.9 to 21.3) | (41.2 to 63.2) | (28.3 to 43.5) | (83.4 to 127.8) | (45.8 to 40.6) |
| | Difference (1–2) | -0.3 | 0.1 | -0.1 | -0.3 | -0.3 |
| | | (-1.6 to 1.2) | (-2.5 to 2.8) | (-2.7 to 2.5) | (-6.7 to 6.3) | (-3.7 to 3.4) |
| | Scenario 1 | 800.6 | 141.9 | 285.1 | 1,228.00 | -15,247.30 |
| | | (631.8 to 960.6) | (112.4 to 170.4) | (225.8 to 342.7) | (970.4 to 1,472.0) | (-15,222.2 to -15,270.9) |
| **Whole population aged 35–100** | Scenario 2 | 788.5 | 140.6 | 277.5 | 1,206.60 | -15,279.40 |
| | | (623.1 to 943.6) | (111.1 to 168.2) | (219.4 to 332.1) | (952.6 to 1,443.0) | (-15,253.9 to -15,304.3) |
| | Difference (1–2) | 12.5 | 1.5 | 7.7 | 21.6 | 32.3 |
| | | (8.9 to 16.7) | (-0.8 to 3.9) | (3.3 to 12.5) | (13.0 to 31.9) | (-3.5 to 69.6) |

£billions in 2019 prices (95% uncertainty intervals in brackets). Comparing Scenario 1 –CVD Plateau, with Scenario 2- CVD Fall.

Note: QALYs are quality adjusted life years. QALYs for CVD and dementia are QALYs lost through disease; QALYs across the whole population is QALYs experienced, so is displayed as a negative value, as it is QALYs lived rather than lost. Data sources same as Table 1.

Our total costs of social care for 35–100 year olds were approximately £14 billion per year. This was slightly lower than the £22 billion quoted for England in 2017/18 [24]. (However, almost half that spend was on people aged under 65 [25]).

Our total cost of informal care for adults aged 35 and over was approximately £28 billion per year for England and Wales, slightly lower than the corresponding ONS household accounts figure of some £55billion; that however included all adults aged 18 and over [26].

## Strengths and weaknesses

Our study uses a single source of data, from a large representative survey of older people in England linked to administrative healthcare records, to estimate both incidence of CVD and dementia, and associated health and social care use. This enhances both precision and internal consistency. The use of administrative data to cost healthcare use also provides more accurate estimates than sample data alone. Conversely, most previous studies relied on a patchwork of data sources. The IMPACT-BAM model accounts for complex epidemiological interactions between CVD, dementia and disability, notably lag times and competing risks.

Our model estimates of dementia and CVD prevalence were previously validated by comparisons with real life data from HES and Cognitive Function and Ageing Studies (CFAS), showing a good level of agreement [12]. Furthermore, our model formally accounts for the uncertainty of input parameters by using rigorous probabilistic sensitivity analyses. Furthermore, being able to quantify excess costs meant that we could isolate the true cost impact of dementia.

This study also has limitations, notably that our cost estimates are likely to be conservative because we a) did not include the effect of healthcare or social care on changing the QALYs

experienced; b) assumed no change in healthcare services efficiency over time (either reduced costs, or increased health per unit of spend), nor changes in costs due to new technologies or inflation; c) based some cost estimates on self-reported resource use rather than direct data collection [27]. Finally, while ELSA is reasonably representative of the non-institutionalised population, it may predict residential care prevalence and costs less well. We have included some one-way sensitivity analyses (value of QALY, discount rate), as well as the probabilistic sensitivity analysis; but more one-way sensitivity analyses may highlight the drivers of differences in outcomes in more detail. We plan to produce a future study to explore these drivers more, using decomposition analysis.

## Implications for policy makers

The Covid-19 pandemic has clearly changed the overall mortality and health and social care costs trajectory for England and Wales. Given that CVD is a risk factor for many conditions and disabilities, including excess risk of covid-19 death [28] it is important for policy makers to prevent as much CVD as possible in the coming decades. If the recent plateau in CVD incidence and mortality is allowed to persist, that could substantially increase healthcare and social care costs, and the opportunity costs of informal care. Conversely, introducing CVD prevention policies of proven effectiveness could continue the historical decline in CVD and decrease healthcare and social care requirements. Such investments would actually be cost-saving [29].

There is ample room for further reductions in CVD incidence and mortality. Populations similar to ours have CVD rates 50% lower (Singapore) and even 80% lower (Qatar) as highlighted by the Global Burden of Disease studies [30]. The recent UK increases in obesity, diabetes and associated hypertension must therefore be considered key targets for a renewed CVD and dementia prevention strategy. Indeed, as 90% of CVD can be explained by dietary and behavioural risk factors [31], the focus on primordial and primary prevention is now more urgently needed than ever [13]. Improving diet by increasing fruit and veg intake, reducing salt and processed food intake, as well as tobacco control interventions, and reducing sedentary behaviour [3], are examples of cost effective interventions to reduce CVD. There are also clinical interventions that can reduce the case fatality ratio such as risk stratification, blood pressure and lipid control, and revascularisation [32]. Persistent policy stasis will likely worsen the already large inequalities in CVD and dementia [33]. It will also add further pressure to our strained healthcare system. Dementia is now the leading cause of death for women in England and Wales and since it shares risk factors with CVD, it may be that future dementia mortality will be even higher than predicted [34, 35].

Our study may help inform resource allocation decisions. Regardless of scenario, investments in social care may need to increase even more quickly than investments in healthcare. We hope our evidence regarding epidemiological shifts and future health and social care costs might also prove useful for policymakers planning greater integration of health and social care.

## Future research

Future research could use our IMPACT-BAM model to look further at health inequalities, and produce more granular estimates at regional or at local authority levels, especially given the increasingly local input to resource allocation [36]. Future modelling studies could estimate the potential of policy and clinical interventions that may change future prevalence of CVD and dementia, for instance around diet, physical activity, or cognitive training, to provide policy makers with comparative options to more comprehensively inform their actions [37].

The excess costs of CVD and dementia by age used here may be of interest to health economists as potential model input parameters. For instance, an intervention to prevent CVD in

people aged 65–79 might save approximately £2,300 per person-year in healthcare costs; or an intervention to prevent dementia in people aged 80–100 might save around £9,900 per person-year in QALY losses (Table 2).

It might also be useful to look further at productivity impacts. A high proportion of the ELSA sample are above typical working age so there may not be significant lost earnings through ill health. However, there are still household productivity impacts, and friends or relatives providing informal care who lose potential earnings. Modelling these dynamic relationships between social care and informal care in more detail, using time-use survey data might be useful.

## Conclusions

Our analysis suggests that the recent slowdown in CVD improvements could generate substantial additional human, health and social care costs over the next decade. Furthermore, social care costs for older adults may grow twice as fast as healthcare costs over the next decade, regardless of future improvements. Living with CVD also means a greater risk of other age-related disabilities.

Though challenging, funding policy for health and social care needs to be urgently addressed. Finally, while addressing the existing burden of CVD, dietary and tobacco control policies to achieve substantially better CVD prevention will need to be intensified.

## Supporting information

**S1 File. Technical appendix.**
(DOCX)

**S2 File. Info on model R markdown file.**
(MD)

**S1 Data. Initial distribution.**
(CSV)

**S2 Data. Initial population size.**
(CSV)

**S3 Data. Population projections.**
(CSV)

**S4 Data. Transition probabilities CVD fall 2006–18.**
(CSV)

**S5 Data. Transition probabilities CVD fall 2019–30.**
(CSV)

**S6 Data. Transition probabilities CVD plateau 2006–18.**
(CSV)

**S7 Data. Transition probabilities CVD plateau 2019–30.**
(CSV)

## Author Contributions

**Conceptualization:** Brendan Collins, Maria Guzman-Castillo, Jonathan Pearson-Stuttard, George Stoye, Jeremy McCauley, Sara Ahmadi-Abhari, Marzieh Araghi, Martin J. Shipley, Simon Capewell, Eric French, Eric J. Brunner, Martin O'Flaherty.

**Data curation:** Brendan Collins, Piotr Bandosz, George Stoye, Jeremy McCauley, Sara Ahmadi-Abhari.

**Formal analysis:** Brendan Collins, Piotr Bandosz, Maria Guzman-Castillo, Jonathan Pearson-Stuttard, George Stoye, Jeremy McCauley, Sara Ahmadi-Abhari, Marzieh Araghi, Martin J. Shipley, Eric French.

**Funding acquisition:** Simon Capewell, Eric J. Brunner, Martin O'Flaherty.

**Investigation:** Brendan Collins, Piotr Bandosz, Maria Guzman-Castillo, Jonathan Pearson-Stuttard, George Stoye, Jeremy McCauley, Sara Ahmadi-Abhari, Marzieh Araghi, Simon Capewell, Eric French, Eric J. Brunner, Martin O'Flaherty.

**Methodology:** Brendan Collins, Maria Guzman-Castillo, Jonathan Pearson-Stuttard, George Stoye, Jeremy McCauley, Sara Ahmadi-Abhari, Marzieh Araghi, Martin J. Shipley, Simon Capewell, Eric French, Eric J. Brunner, Martin O'Flaherty.

**Project administration:** Brendan Collins, Eric J. Brunner, Martin O'Flaherty.

**Supervision:** Brendan Collins, Eric French, Martin O'Flaherty.

**Visualization:** Piotr Bandosz, Sara Ahmadi-Abhari.

**Writing – original draft:** Brendan Collins, Piotr Bandosz, Maria Guzman-Castillo, Jonathan Pearson-Stuttard, George Stoye, Jeremy McCauley, Sara Ahmadi-Abhari, Marzieh Araghi, Martin J. Shipley, Simon Capewell, Eric French, Eric J. Brunner, Martin O'Flaherty.

**Writing – review & editing:** Brendan Collins, Piotr Bandosz, Jonathan Pearson-Stuttard, George Stoye, Jeremy McCauley, Sara Ahmadi-Abhari, Marzieh Araghi, Martin J. Shipley, Simon Capewell, Eric French, Eric J. Brunner, Martin O'Flaherty.

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
