## [Decision Letter · Decision Letter 0]

5 Nov 2021

PONE-D-21-22992What will the cardiovascular disease slowdown cost?  Modelling the impact of CVD trends on dementia, disability, and economic costs in England and Wales from 2020-2029.PLOS ONE

Dear Dr. Collins,

Thank you for submitting your manuscript to PLOS ONE. After careful consideration, we feel that it has merit but does not fully meet PLOS ONE’s publication criteria as it currently stands. Therefore, we invite you to submit a revised version of the manuscript that addresses the points raised during the review process.

To better appreciate your interesting analysis, could you please particularly pay attention to the following excellent points raised by the two reviewers: 1) Please include few paragraphs in the methods section that summarize with sufficient amount of detail how probabilities were defined for transitioning across health states. It is well understood the model simulates an open cohort aged 35-100 over 10 year time horizon. However it is unclear to the reader how new cohorts enter the model (those that turn 35, and would this be on an annual basis), what the initial distributions were for each health state, and how transitions were modeled from disease-free states to disease/death states and from disease to death states (e.g., were transitions based on age and other demographic variables)?  2) Please include a Table 1 summarizing the model parameters and distributions used for transition probabilities, costs and utility weights together with data sources. When a regression equation was used, could you please state this in the table and refer to its source? 

3) Please justify the beta distribution type and OLS for modeling costs. Describe also here how uncertainty was modeled within PSA. Which parameters contributed to the uncertainty and which were assumed to be fixed? How was correlation between parameters incorporated?

4) Please include more details about the 2 modeled scenarios as indicated by Reviewer 1 and what the implications of assumptions are in the model. Also please clarify whether and which costs depend on gains in life expectancy.

5) Better justify use/non-use of discounting, time horizon and monetary WTP value for a QALY. 

6) Please add one-way sensitivity analyses of key parameters as suggested by Reviewer 2.

In addition, several more textual suggestions were made by both reviewers.

We look forward to receiving your revised manuscript.

Kind regards,

Bart Ferket

Academic Editor

PLOS ONE

Journal Requirements:

2. Please amend the manuscript submission data (via Edit Submission) to include authors: Piotr Bandosz, Maria Guzman-Castillo, Jonathan Pearson-Stuttard, George Stoye, Jeremy McCauley, Sara Ahmadi-Abhari, Marzieh Araghi, Martin J Shipley, Simon Capewell, Eric French, Eric J Brunne, Martin O’Flaherty

3. Thank you for stating the following in the Competing Interests/Financial Disclosure section:

“All authors have completed the Unified Competing Interest form (available on request from the corresponding author) and declare: no support from any organisation for the submitted work; no financial relationships with any organisations that might have an interest in the submitted work in the previous three years. Dr Collins is secondment as Head of Health Economics in Welsh Government, this paper does not represent any views of Welsh Government. Dr Pearson-Stuttard is also Head of Health Analytics at Lane Clark & Peacock LLP, vice-chair of the Royal Society for Public Health and reports personal fees from Novo Nordisk A/S, all outside of the submitted work.”

We note that one or more of the authors are employed by a commercial company:Health Analytics at Lane Clark & Peacock LLP

Reviewers' comments:

Reviewer's Responses to Questions

**Comments to the Author**

1. Is the manuscript technically sound, and do the data support the conclusions?

Reviewer #1: Partly

Reviewer #2: Yes

2. Has the statistical analysis been performed appropriately and rigorously? 

Reviewer #1: Yes

Reviewer #2: Yes

3. Have the authors made all data underlying the findings in their manuscript fully available?

Reviewer #1: No

Reviewer #2: No

4. Is the manuscript presented in an intelligible fashion and written in standard English?

Reviewer #1: No

Reviewer #2: Yes

5. Review Comments to the Author

Reviewer #1: This study is a model-based analysis of the economic implications of the recent trend in CVD incidence in England and Wales compared to a counterfactual of persisting long-term trend. Using a probabilistic Markov model (IMPACT-BAM), the study simulated and compared two scenarios: 1) CVD Plateau, where CVD incidence remains at 2011 levels, and 2) CVD Fall, where CVD incidence follows the long-term trends and continues to decline. It concludes that the slowdown in the decline of CVD incidence is associated with substantial costs.

The topic is potentially interesting, and the findings could help guide priority setting in the local setting. Unfortunately, key methodological details were not sufficiently described in the text to determine the validity of the study. I have included some major issues mostly around the methods and other comments and questions in the following section.

Major issues:

1. The methods section lacks essential details. I understand that the study used a previously published simulation model, but it was unclear how the two scenarios were implemented exactly in this particular study. For example, how was the trend of CVD incidence in the two scenarios fitted statistically? What is the risk profile of those who develop CVD in scenario 1 compared to scenario 2? Considering that the comparison between these two scenarios is the central question of this study, it was very difficult to determine its methodological validity.

2. It was also unclear how CVD and dementia were correlated in this analysis, i.e., how a change in CVD incidence trend between two scenarios affects the health and cost outcomes of dementia, and how this effect was operationalized in the simulation model.

3. A decrease in CVD incidence is associated with longer life expectancy, but this prolonged life expectancy also comes with some future unrelated healthcare costs: if we prevent individuals from developing CVD, they may still seek care and incur health spending because of other conditions. Not accounting for such costs may bias the results in favor of the CVD Fall scenario. I could not find whether (and if yes, how) this study includes such costs from the description of the methods.

4. Since this study projects into the next 10 years to calculate the economic impact of CVD trends, one would think that discounting is crucial to account for time preferences. However, it was not clear why discounting was not performed, and I think it would help to see both discounted and undiscounted results.

5. The manuscript requires significant copyediting. References are sometimes misaligned (e.g., Introduction -> paragraph 3 -> line 3, the ELSA study is referenced to #8, but should be #11 instead). Many sections/paragraphs lack structure and focus -- the “Implications for policy makers” section is a prime example consisting of scattered paragraphs of just 1-2 sentences; they should be combined to indicate a clear flow of logic. Another example: the “Social care and informal care costs” section includes some methodological descriptions of the QALYs, which should belong to the next section. Some abbreviations are used before they are defined (e.g., “HES”). Additionally, there are grammar errors (incorrect punctuations and capitalizations, etc.) throughout the manuscript.

Other comments:

1. The study uses £60,000/QALY as the threshold to value QALYs and references the UK Treasury Green Book. However, the most often used threshold we see from the UK is £30,000/QALY recommended by NICE, substantially different from £60,000/QALY. So, what is the rationale for using £60,000/QALY? I also wonder how the results would change if alternative thresholds were used.

2. Why were costs fitted to beta distributions? Beta distributions are more commonly used for probabilities and utility weights in probabilistic sensitivity analysis, and Gamma or Log-normal are more appropriate for costs.

3. I’m having a hard time understanding what this sentence means: “The QALYs reflect only the uncertainty in the epidemiology, not uncertainty around the QALY impact of disease, which is reflected in very tight confidence intervals.” Is this essentially saying that there’s no probabilistic distribution added to the QALY weights because the uncertainty on QALY weights is very low? In any case, this sentence should be reworded to improve clarity.

4. Tables 1 and 2: These tables present results for a “base case scenario.” It’s confusing because a base case scenario is not described and defined in the methods section.

5. Scenario 2 is a hypothetical scenario where CVD incidence declines at a rate close to historical levels. Even though the discussion section touches briefly on the aspects CVD prevention could focus on, scenario 2 seems like a very hypothetical and unrealistic scenario without some justification on *how* this decline could be achieved. More and deeper discussion on this could be helpful, and maybe a somewhat more realistic scenario could be of more policy interest.

6. The “Future research” section claims that “Future research could use our IMPACT-BAM model to look further at health inequalities, and produce more granular estimates at regional or at local authority levels, especially given the increasingly local input to resource allocation.” It was unclear, though, whether the authors have made the IMPACT-BAM model publicly available for such uses. I could not find a user interface for the model from an online search.

Reviewer #2: This is a well-conducted and important study that quantifies the costs and QALYs associated with persistent plateauing versus continued decline of cardiovascular disease rates in England and Wales.

Below I provide some comments on each section of the abstract. While most of these are minor recommendations, I strongly recommend that the simulation procedure is more clearly explained in the Methods section and that sensitivity analysis is conducted to better quantify the contribution of individual model parameters to overall uncertainty.

Abstract

The abstract provides a detailed description of the study.

Introduction

The Introduction details declining CVD mortality rates during the late twentieth century and their subsequent plateauing. The respective causes of declines in CVD and CHD mortality in the U.K. have been discussed in prior literature (e.g., Bajekal et al., PLoS Med, 2012 and O’Flaherty, Buchan, and Capewell, Heart, 2012). Some discussion of the relative impact of different risk factor exposures and novel treatments would be informative.

The Introduction clearly sets out the important relationship between CVD and dementia, explains the need for a study to estimate future health and cost consequences associated with the ‘cardiovascular disease slowdown’, and summarises how this will be achieved.

Methods

The methods are well-described and are appropriate for the research question, employing a previously validated Markov model. Previous publications which employed the model should be cited when the IMPACT BAM model is introduced in the Methods section. Further information on the basis for the U.K. treasury’s decision to value QALYs at £60,000 may provide context to this model parameter (i.e., determined based on revealed preference studies which aim to quantify the statistical value of a life). A short justification for the study time horizon would also be helpful.

Generally, the Methods section would be improved with a subsection which clearly describes the simulation approach. The manuscript refers to ‘probabilistic sensitivity analysis’ once in the Methods and once in the Strengths and Limitations sections. It would be useful to have a short section of the Methods which describes how cohorts transition through the model (including a model figure, as recommended by Consolidated Health Economic Evaluation Reporting Standards guideline), how many cycles are included in each run, the process by which parameters are stochastically sampled, the approach to defining probability distributions for each of the model parameters, and the way summary statistics from probabilistic runs are reported. I would strongly advise inclusion of a ‘Table 1’ which describes all the model inputs, and their mean values, ranges, distributions for probabilistic analysis, and sources. The suggested figure and table could be included in the supplement material.

The probabilistic simulation approach helps to quantify the uncertainty inherent in this modelling study. The contribution of individual model parameters to this uncertainty is unclear. I recommend conducting ‘traditional’ sensitivity analysis, whereby most parameters are held constant while some parameters are systematically varied (either deterministically or probabilistically). The impact of these results on important outcomes (e.g., incremental health, social and care and total costs) could be presented in tornado diagrams.

Results

The results section is well-written, containing all relevant results. The number of QALYs and life years accumulated in both scenarios are important intermediate outcomes that could also be reported.

Discussion

The discussion provides a useful summary of the results, sets the manuscript in the context of similar literature, and outlines strengths and limitations. The section ‘Implications for Policy Makers’ may be improved by citing examples of cost-effective cardiovascular prevention policies which could be implemented by policy-makers to arrest the plateauing of CVD rates.

6. PLOS authors have the option to publish the peer review history of their article (what does this mean?). If published, this will include your full peer review and any attached files.

Reviewer #1: No

Reviewer #2: **Yes: **Ciaran Kohli-Lynch

---

## [Author Response · Author response to Decision Letter 0]

11 Mar 2022

Response to reviewers

Thank you for submitting your manuscript to PLOS ONE. After careful consideration, we feel that it has merit but does not fully meet PLOS ONE’s publication criteria as it currently stands. Therefore, we invite you to submit a revised version of the manuscript that addresses the points raised during the review process.

To better appreciate your interesting analysis, could you please particularly pay attention to the following excellent points raised by the two reviewers:

1) Please include few paragraphs in the methods section that summarize with sufficient amount of detail how probabilities were defined for transitioning across health states. It is well understood the model simulates an open cohort aged 35-100 over 10 year time horizon. However it is unclear to the reader how new cohorts enter the model (those that turn 35, and would this be on an annual basis), what the initial distributions were for each health state, and how transitions were modelled from disease-free states to disease/death states and from disease to death states (e.g., were transitions based on age and other demographic variables)? 

Thank you. We have added some paragraphs to describe this (p.4) and have added a more description in the Appendix detailing how transition probabilities were calculated. 

We have also added a new Table 1 with model inputs, as requested below. 

We have also attached a file with full model inputs. Model inputs vary by single year of age and calendar year so would be very bulky to include in the main body of the paper. 

We have added more clarity around the new cohort of 35 year olds: A new cohort of those reaching age 35 each year enters through the disease free state as displayed in the arrow in the Model figures. Prevalence of cardiovascular disease and functional impairment is very low in this cohort (<2% in total). Therefore, the resulting error in misclassification is negligible. 

2) Please include a Table 1 summarizing the model parameters and distributions used for transition probabilities, costs and utility weights together with data sources. When a regression equation was used, could you please state this in the table and refer to its source? 

Thank you. We have added a file with all of the model inputs. We have added a new Table 1. Most of the regression equations are quite complex (e.g. lots of age groups); however, all are now included in the Appendix. 

3) Please justify the beta distribution type and OLS for modeling costs. Describe also here how uncertainty was modelled within PSA. Which parameters contributed to the uncertainty and which were assumed to be fixed? How was correlation between parameters incorporated?

Thank you. The costs were modelled with a two stage least squares regression for the relationship between cost, health state, gender and age. The overall costs were fitted to a beta around +/- 20% to account for other uncertainties that were not accounted for. We used beta based on this paper https://www.york.ac.uk/media/economics/documents/herc/wp/11_31.pdf - and to account for healthcare costs typically being right-skewed in shape.

4) Please include more details about the 2 modelled scenarios as indicated by Reviewer 1 and what the implications of assumptions are in the model. Also please clarify whether and which costs depend on gains in life expectancy.

Thank you. We have added more detail of the 2 modelled scenarios and justification for these scenarios. 

5) Better justify use/non-use of discounting, time horizon and monetary WTP value for a QALY. 

Have added discounted results (using 3.5% for costs and 1.5% QALYs in line with UK Treasury) and £30k valuation of QALYs in Appendix 3.

6) Please add one-way sensitivity analyses of key parameters as suggested by Reviewer 2.

We agree that one-way sensitivity analyses, often shown in the form of tornado charts are really useful in showing the key sources of uncertainty around incremental net monetary benefit between two or more interventions. However, this paper is not a traditional cost effectiveness analysis comparing an intervention with a comparator. It is comparing two scenarios where the sources of uncertainty are generally the same for both scenarios so would cancel each other out somewhat in a tornado diagram. We are planning to carry out more work in the future looking at the drivers in healthcare costs in different model scenarios using blinder-oaxaca decomposition but we feel it is beyond the scope of the present paper which is quite lengthy already.

In addition, several more textual suggestions were made by both reviewers.

We look forward to receiving your revised manuscript.

Kind regards,

Bart Ferket

Academic Editor

PLOS ONE

Journal Requirements:

2. Please amend the manuscript submission data (via Edit Submission) to include authors: Piotr Bandosz, Maria Guzman-Castillo, Jonathan Pearson-Stuttard, George Stoye, Jeremy McCauley, Sara Ahmadi-Abhari, Marzieh Araghi, Martin J Shipley, Simon Capewell, Eric French, Eric J Brunne, Martin O’Flaherty

3. Thank you for stating the following in the Competing Interests/Financial Disclosure section:

“All authors have completed the Unified Competing Interest form (available on request from the corresponding author) and declare: no support from any organisation for the submitted work; no financial relationships with any organisations that might have an interest in the submitted work in the previous three years. Dr Collins is secondment as Head of Health Economics in Welsh Government, this paper does not represent any views of Welsh Government. Dr Pearson-Stuttard is also Head of Health Analytics at Lane Clark & Peacock LLP, vice-chair of the Royal Society for Public Health and reports personal fees from Novo Nordisk A/S, all outside of the submitted work.”

We note that one or more of the authors are employed by a commercial company:Health Analytics at Lane Clark & Peacock LLP

Thanks have added this about the funder BHF.

Thank you. We have now added this.

Reviewers' comments:

Reviewer's Responses to Questions

Comments to the Author

1. Is the manuscript technically sound, and do the data support the conclusions?

Reviewer #1: Partly

Reviewer #2: Yes

2. Has the statistical analysis been performed appropriately and rigorously?

Reviewer #1: Yes

Reviewer #2: Yes

3. Have the authors made all data underlying the findings in their manuscript fully available?

We have included a file with a detailed set of model inputs. We plan to publish the complete model at some point in the future, but need to work through it with funders and authors. 

Reviewer #1: No

Reviewer #2: No

4. Is the manuscript presented in an intelligible fashion and written in standard English?

Reviewer #1: No

Reviewer #2: Yes

5. Review Comments to the Author

Reviewer #1: This study is a model-based analysis of the economic implications of the recent trend in CVD incidence in England and Wales compared to a counterfactual of persisting long-term trend. Using a probabilistic Markov model (IMPACT-BAM), the study simulated and compared two scenarios: 1) CVD Plateau, where CVD incidence remains at 2011 levels, and 2) CVD Fall, where CVD incidence follows the long-term trends and continues to decline. It concludes that the slowdown in the decline of CVD incidence is associated with substantial costs.

The topic is potentially interesting, and the findings could help guide priority setting in the local setting. Unfortunately, key methodological details were not sufficiently described in the text to determine the validity of the study. I have included some major issues mostly around the methods and other comments and questions in the following section.

Major issues:

1. The methods section lacks essential details. I understand that the study used a previously published simulation model, but it was unclear how the two scenarios were implemented exactly in this particular study. For example, how was the trend of CVD incidence in the two scenarios fitted statistically? What is the risk profile of those who develop CVD in scenario 1 compared to scenario 2? Considering that the comparison between these two scenarios is the central question of this study, it was very difficult to determine its methodological validity.

Thank you, we have added more detail for the methods and for the two scenarios to the paper and the appendix.

2. It was also unclear how CVD and dementia were correlated in this analysis, i.e., how a change in CVD incidence trend between two scenarios affects the health and cost outcomes of dementia, and how this effect was operationalized in the simulation model.

Thank you. The model is built on ELSA data where having CVD increases the probability of transition to dementia vs. a no dementia state. We have included more detail of this in the paper and appendix.

3. A decrease in CVD incidence is associated with longer life expectancy, but this prolonged life expectancy also comes with some future unrelated healthcare costs: if we prevent individuals from developing CVD, they may still seek care and incur health spending because of other conditions. Not accounting for such costs may bias the results in favor of the CVD Fall scenario. I could not find whether (and if yes, how) this study includes such costs from the description of the methods.

Thank you. The study includes future unrelated healthcare costs. We have made this more clear in the description. This is one of the key questions we aim to answer: does a CVD reduction save money and QALYs even if it means people live longer. The answer suggested by our study is yes.

4. Since this study projects into the next 10 years to calculate the economic impact of CVD trends, one would think that discounting is crucial to account for time preferences. However, it was not clear why discounting was not performed, and I think it would help to see both discounted and undiscounted results.

Thank you. We have added a discounted sensitivity analysis – at 3.5% for costs, 1.5% for QALYs, , in line with NICE and UK Treasury. Discounting was not initially carried out because we are not looking at a specific intervention and wanted to give an idea of how health system costs will evolve year on year, which discounting may partially obscure. 

5. The manuscript requires significant copyediting. References are sometimes misaligned (e.g., Introduction -> paragraph 3 -> line 3, the ELSA study is referenced to #8, but should be #11 instead). Many sections/paragraphs lack structure and focus -- the “Implications for policy makers” section is a prime example consisting of scattered paragraphs of just 1-2 sentences; they should be combined to indicate a clear flow of logic. Another example: the “Social care and informal care costs” section includes some methodological descriptions of the QALYs, which should belong to the next section. Some abbreviations are used before they are defined (e.g., “HES”). Additionally, there are grammar errors (incorrect punctuations and capitalizations, etc.) throughout the manuscript.

Thank you, we have corrected these examples and have conducted a further round of copy editing. 

Other comments:

1. The study uses £60,000/QALY as the threshold to value QALYs and references the UK Treasury Green Book. However, the most often used threshold we see from the UK is £30,000/QALY recommended by NICE, substantially different from £60,000/QALY. So, what is the rationale for using £60,000/QALY? I also wonder how the results would change if alternative thresholds were used.

Thank you. We have added a sensitivity analysis using £30k per QALY. The NICE threshold is used for recommending technologies for the NHS which has a (typically) fixed budget. The UK Treasury valuation is used for considering policy interventions which may be more appropriate for the present study. 

2. Why were costs fitted to beta distributions? Beta distributions are more commonly used for probabilities and utility weights in probabilistic sensitivity analysis, and Gamma or Log-normal are more appropriate for costs.

Thank you. We used beta based on this paper which suggests GB2 can be the best distribution for healthcare costs which are often right-skewed https://www.york.ac.uk/media/economics/documents/herc/wp/11_31.pdf

3. I’m having a hard time understanding what this sentence means: “The QALYs reflect only the uncertainty in the epidemiology, not uncertainty around the QALY impact of disease, which is reflected in very tight confidence intervals.” Is this essentially saying that there’s no probabilistic distribution added to the QALY weights because the uncertainty on QALY weights is very low? In any case, this sentence should be reworded to improve clarity.

Thank you, yes. We have reworded it as you suggest to make it clearer. 

4. Tables 1 and 2: These tables present results for a “base case scenario.” It’s confusing because a base case scenario is not described and defined in the methods section.

Thank you for picking this up. We have changed this – it is actually Scenario 2, although the results for this are very similar for both scenarios. 

5. Scenario 2 is a hypothetical scenario where CVD incidence declines at a rate close to historical levels. Even though the discussion section touches briefly on the aspects CVD prevention could focus on, scenario 2 seems like a very hypothetical and unrealistic scenario without some justification on *how* this decline could be achieved. More and deeper discussion on this could be helpful, and maybe a somewhat more realistic scenario could be of more policy interest.

Thank you. We give some examples of countries that have continued to show reductions in CVD incidence and that have much lower incidence than the UK so we do not believe the scenario we are presenting is unrealistic, however it may take a longer time to achieve, especially given the pandemic. 

6. The “Future research” section claims that “Future research could use our IMPACT-BAM model to look further at health inequalities, and produce more granular estimates at regional or at local authority levels, especially given the increasingly local input to resource allocation.” It was unclear, though, whether the authors have made the IMPACT-BAM model publicly available for such uses. I could not find a user interface for the model from an online search.

Thank you. We aim to make the IMPACT-BAM model available in due course. Furthermore, we have a big group working on the model, so some future research may be carried out by people with previous experience of working with the model. 

Reviewer #2: This is a well-conducted and important study that quantifies the costs and QALYs associated with persistent plateauing versus continued decline of cardiovascular disease rates in England and Wales.

Below I provide some comments on each section of the abstract. While most of these are minor recommendations, I strongly recommend that the simulation procedure is more clearly explained in the Methods section and that sensitivity analysis is conducted to better quantify the contribution of individual model parameters to overall uncertainty.

Abstract

The abstract provides a detailed description of the study.

Introduction

The Introduction details declining CVD mortality rates during the late twentieth century and their subsequent plateauing. The respective causes of declines in CVD and CHD mortality in the U.K. have been discussed in prior literature (e.g., Bajekal et al., PLoS Med, 2012 and O’Flaherty, Buchan, and Capewell, Heart, 2012). Some discussion of the relative impact of different risk factor exposures and novel treatments would be informative.

Thank you. We have added some reference to treatments that may reduce the CVD case fatality ratio; we are mainly concerned with CVD incidence in this study related to policy interventions, rather than tertiary prevention but agree tertiary prevention is important. 

The Introduction clearly sets out the important relationship between CVD and dementia, explains the need for a study to estimate future health and cost consequences associated with the ‘cardiovascular disease slowdown’, and summarises how this will be achieved.

Methods

The methods are well-described and are appropriate for the research question, employing a previously validated Markov model. Previous publications which employed the model should be cited when the IMPACT BAM model is introduced in the Methods section. Further information on the basis for the U.K. treasury’s decision to value QALYs at £60,000 may provide context to this model parameter (i.e., determined based on revealed preference studies which aim to quantify the statistical value of a life). A short justification for the study time horizon would also be helpful.

Thank you. We have added reference to the study time horizon mirroring the NHS plan. We have also added sensitivity analysis with £30k / QALY valuation.

Generally, the Methods section would be improved with a subsection which clearly describes the simulation approach. The manuscript refers to ‘probabilistic sensitivity analysis’ once in the Methods and once in the Strengths and Limitations sections. It would be useful to have a short section of the Methods which describes how cohorts transition through the model (including a model figure, as recommended by Consolidated Health Economic Evaluation Reporting Standards guideline), how many cycles are included in each run, the process by which parameters are stochastically sampled, the approach to defining probability distributions for each of the model parameters, and the way summary statistics from probabilistic runs are reported. I would strongly advise inclusion of a ‘Table 1’ which describes all the model inputs, and their mean values, ranges, distributions for probabilistic analysis, and sources. The suggested figure and table could be included in the supplement material.

Thank you. We have added a Table 1, and model figure into the Appendix.

The probabilistic simulation approach helps to quantify the uncertainty inherent in this modelling study. The contribution of individual model parameters to this uncertainty is unclear. I recommend conducting ‘traditional’ sensitivity analysis, whereby most parameters are held constant while some parameters are systematically varied (either deterministically or probabilistically). The impact of these results on important outcomes (e.g., incremental health, social and care and total costs) could be presented in tornado diagrams.

Thank you. Please see comment above. We aim to carry out future work looking in more detail at the drivers of change and uncertainty in the model.

Results

The results section is well-written, containing all relevant results. The number of QALYs and life years accumulated in both scenarios are important intermediate outcomes that could also be reported.

Thank you. We have made more reference to the incremental QALY results. 

Discussion

The discussion provides a useful summary of the results, sets the manuscript in the context of similar literature, and outlines strengths and limitations. The section ‘Implications for Policy Makers’ may be improved by citing examples of cost-effective cardiovascular prevention policies which could be implemented by policy-makers to arrest the plateauing of CVD rates.

Thank you. We have added more reference to cost effective prevention policies in this section. 

Huge thanks to both reviewers and to the editors for providing such perspicacious comments. We hope that we have acted upon them to your satisfaction, and that our paper is now fit for publication in PLOS ONE.

---

## [Decision Letter · Decision Letter 1]

27 Apr 2022

PONE-D-21-22992R1What will the cardiovascular disease slowdown cost?  Modelling the impact of CVD trends on dementia, disability, and economic costs in England and Wales from 2020-2029.PLOS ONE

Dear Dr. Collins,

Thank you for submitting your manuscript to PLOS ONE. After careful consideration, we feel that it has merit but does not fully meet PLOS ONE’s publication criteria as it currently stands. Therefore, we invite you to submit a revised version of the manuscript that addresses the points raised during the review process. Please perform the additional one-way sensitivity analyses as suggested or provide a better justification for not performing these in the discussion (strengths and weaknesses) section in response to the remaining concerns of the reviewer. Note that the sentence "Hence the value of our extensive sensitivity analyses." may not be supported well by your analyses in the current version based on the comments of the reviewer.

We look forward to receiving your revised manuscript.

Kind regards,

Bart Ferket

Academic Editor

PLOS ONE

Journal Requirements:

Reviewers' comments:

Reviewer's Responses to Questions

**Comments to the Author**

1. If the authors have adequately addressed your comments raised in a previous round of review and you feel that this manuscript is now acceptable for publication, you may indicate that here to bypass the “Comments to the Author” section, enter your conflict of interest statement in the “Confidential to Editor” section, and submit your "Accept" recommendation.

Reviewer #2: (No Response)

2. Is the manuscript technically sound, and do the data support the conclusions?

Reviewer #2: Yes

3. Has the statistical analysis been performed appropriately and rigorously? 

Reviewer #2: Yes

4. Have the authors made all data underlying the findings in their manuscript fully available?

Reviewer #2: Yes

5. Is the manuscript presented in an intelligible fashion and written in standard English?

Reviewer #2: Yes

6. Review Comments to the Author

Reviewer #2: I am generally satisfied that the authors have responded to reviewer comments, especially regarding improved descriptions of the modelling process in the Methods section.

I previously suggested that one-way sensitivity analysis could be conducted to better explore uncertainty in the estimates arttributable to individual model parameters. The authors responded that,

"...this paper is not a traditional cost effectiveness analysis comparing an intervention with a comparator. It is comparing two scenarios where the sources of uncertainty are generally the same for both scenarios so would cancel each other out somewhat in a tornado diagram."

I still believe that varying key parameters (i.e., those presented in Table 1) will independently impact model outcomes. The contribution of these parameters to the overall uncertainty in the modelling process is useful and important information which helps to both validate the modelling procedure and explore the impact of different cost drivers. My concern regarding the lack of one-way sensitivty analysis remains and I think these analyses should be conducted (whether the results are presented in a tornado diagram or otherwise).

7. PLOS authors have the option to publish the peer review history of their article (what does this mean?). If published, this will include your full peer review and any attached files.

Reviewer #2: **Yes: **Ciaran Kohli-Lynch

---

## [Author Response · Author response to Decision Letter 1]

27 Apr 2022

Response to Reviewers

Dear PLOS One

Thank you for reviewing our study again. Please find rebuttal below;

Please perform the additional one-way sensitivity analyses as suggested or provide a better justification for not performing these in the discussion (strengths and weaknesses) section in response to the remaining concerns of the reviewer. Note that the sentence "Hence the value of our extensive sensitivity analyses." may not be supported well by your analyses in the current version based on the comments of the reviewer.

Thank you for this comment. We have taken out this sentence and replaced it with sentences below. We have included some one-way sensitivity analyses as suggested by reviewers. We have included probabilistic sensitivity analysis which is generally seen as the gold standard in terms of sensitivity analysis but the way the model is structured makes it more difficult to perform a series of one-way sensitivity analyses, but we agree that this would be useful and it is something we intend to do in future studies with this model, to explore the drivers of change over time. So we agree with the reviewer that this will be useful, but we want to explore it more in a potential future study.

“We have included some one-way sensitivity analyses (value of QALY, discount rate), as well as the probabilistic sensitivity analysis; but more one-way sensitivity analyses may highlight the drivers of differences in outcomes in more detail. We plan to produce a future study to explore these drivers more, using decomposition analysis.”

I hope this change in emphasis as well as explaining our future plans answers this to your satisfaction.

With best wishes

Dr Brendan Collins

---

## [Editor Report · Decision Letter 2]

9 May 2022

What will the cardiovascular disease slowdown cost?  Modelling the impact of CVD trends on dementia, disability, and economic costs in England and Wales from 2020-2029.

PONE-D-21-22992R2

Dear Dr. Collins,

We’re pleased to inform you that your manuscript has been judged scientifically suitable for publication and will be formally accepted for publication once it meets all outstanding technical requirements.

Kind regards,

Bart Ferket

Academic Editor

PLOS ONE
---

## [Editor Report · Acceptance letter]

12 May 2022

PONE-D-21-22992R2 

What will the cardiovascular disease slowdown cost?  Modelling the impact of CVD trends on dementia, disability, and economic costs in England and Wales from 2020-2029. 

Dear Dr. Collins:

I'm pleased to inform you that your manuscript has been deemed suitable for publication in PLOS ONE. Congratulations! Your manuscript is now with our production department. 

Kind regards, 

on behalf of

Dr. Bart Ferket 

Academic Editor

PLOS ONE